

# Altered expression of miRNA profile in peripheral blood mononuclear cells following the third dose of inactivated COVID-19 vaccine

Guanguan Qiu[1,*], Ruoyang Zhang[2,*], Huifeng Qian[1],
Ruoqiong Huang[2], Jie Xia[2], Ruoxi Zang[2], Zhenkai Le[2], Qiang Shu[2],
Jianguo Xu[1,2], Guoping Zheng[1] and Jiangmei Wang[2]

[1] Shaoxing Second Hospital, Shaoxing, Zhejiang, China
[2] Children's Hospital of Zhejiang University School of Medicine, Hangzhou, China
* These authors contributed equally to this work.

Corresponding authors
Guoping Zheng, zgp28037@163.com
Jiangmei Wang,
wjmerbao@zju.edu.cn

## ABSTRACT

COVID-19 vaccination is the most effective strategy for preventing severe disease and death. Inactivated vaccines are the most accessible type of COVID-19 vaccines in developing countries. Several studies, including work from our group, have demonstrated that the third dose (booster vaccination) of inactivated COVID-19 vaccine induces robust humoral and cellular immune responses. The present study aimed to examine miRNA expression profile in participants who received a homologous third dose of the CoronaVac vaccine. Samples of peripheral blood mononuclear cells (PBMCs) were collected from healthcare volunteers both before and 1–2 weeks after the booster dose. miRNA microarray analysis in a discovery cohort of six volunteers identified 67 miRNAs with differential expression. Subsequently, the expression of six miRNAs related to immune responses was examined in a validation cohort of 31 participants *via* qRT-PCR. Our results validated the differential expression of miR-25-5p, miR-34c-3p, and miR-206 post-booster, with a significant correlation to the receptor binding domain (RBD)-specific antibody. Bioinformatic analysis suggested that miR-25-5p, miR-34c-3p, and miR-206 may target multiple pathways involved in immune regulation and inflammation. Therefore, our study highlights miR-25-5p, miR-34c-3p, and miR-206 in PBMCs as promising biomarkers for assessing the immune response induced by the booster dose of the CoronaVac vaccine.

## INTRODUCTION

There are two main categories of vaccines for severe acute respiratory syndrome coronavirus 2 (SARS-CoV-2). The first category comprises antigens from viral particles or subunits, such as inactivated virus and protein subunit platforms. The second category uses genetic materials embedded in a specific nucleotide sequence to synthesize SARS-CoV-2 antigens, such as mRNA and adenoviral vector platforms (*Heinz, 2021*). Inactivated

viral vaccines have a good track record for safety and efficacy in the prevention of influenza, rabies, and polio (*Hajj Hussein et al., 2015*). Vaccines in this class are generated by killing the pathogen through physical or chemical procedures. Inactivated vaccines are unable to replicate in vaccinated individuals, but still provoke a protective immune response against future disease (*Chauhan et al., 2021*). Inactivated COVID-19 vaccines, including Covaxin (India), Covilo (China), and CoronaVac (China) are highly effective in preventing severe illness with milder side effects (*Dadras et al., 2022*).

microRNAs (miRNAs) are a class of non-coding single-stranded 18–25 nucleotide RNAs, which regulate up to 60% of protein encoding genes in mammalian cells post-transcriptionally (*Friedman et al., 2009*). miRNA genes are often found in introns of protein-encoding genes and in intergenic regions. Expression of miRNAs can be coregulated by their host genes and their own promoters (*Olena & Patton, 2010*). miRNAs bind with target mRNAs and regulate gene expression *via* translational repression and mRNA decay (*Bartel, 2018*). There are approximately 2,300 true human mature miRNAs with tissue and cell specific expression (*Alles et al., 2019*). An individual miRNA is believed to target various genes, and one functional gene can be regulated by multiple miRNAs (*Xu et al., 2020*).

A number of studies have reported that miRNAs modulate immune response after vaccination. In humans vaccinated with the rVSVΔG-ZEBOV-GP Ebola vaccine, the blood miRNA profile at early time points post-vaccination was correlated with the ZEBOV-specific IgG response at 1 month and 1 year. An eight-miRNA signature was linked to immune-related mRNA targets and pathways (*Vianello et al., 2023*). *Nakaya et al. (2015)* documented that the interferon response was regulated by miR-424 after influenza vaccination. *Miyashita et al. (2019)* showed that miR-451a expression in serum extracellular vesicles (EVs) was inversely associated with the number of local symptoms after seasonal flu vaccination. *Haralambieva et al. (2018)* demonstrated that differential miRNA expression in B cells was correlated with neutralizing antibody titers after measles vaccination. *Fukuyama et al. (2015)* found that nasal nanogel-based pneumococcal vaccine induced the expression of miR-181a and miR-326 in serum and tissue of respiratory tract as well as humoral and cellular immune responses in macaques. *Xiong et al. (2013)* documented that serum miR-155 levels were associated with non-responsiveness to hepatitis B vaccine.

A number of studies have reported that miRNAs modulate immune response during COVID-19 infection. *Sabbatinelli et al. (2021)* found that lower serum levels of miR-146a were associated with a lack of clinical response to tocilizumab, an anti-IL-6 receptor antibody, in COVID-19 patients. *Morales et al. (2022)* reported that levels of miR-223-3p in mouse lungs were upregulated during the SARS-CoV-2 infection. The expression of NLRP3 inflammasome was significantly elevated after inhibition of miRNA-223-3p, suggesting its role in restraining excessive inflammatory response. *Soltani-Zangbar et al. (2023)* documented that SARS-CoV-2 infection promoted the expression of miR-155, which disrupted the equilibrium of Th17/Treg by modulating SOCS1 levels. In lung biopsy samples of patients died of COVID-19, the levels of miR-26a-5p, miR-29b-3p and miR-34a-5p were significantly diminished compared to those of heathy controls. The

expression of miR-26a-5p was significantly correlated with proinflammatory IL-6 and ICAM-1, highlighting its role in mediating the immune response. Additionally, miR-29b-3p had a strong association with the anti-inflammatory cytokines IL-4 and IL-8 (*Centa et al., 2021*). In COVID-19 patients, plasma levels of miR-146a, miR-155, and miR-221 were upregulated compared to the healthy controls. Specifically, miR-146a showed a positive correction with absolute neutrophil count, while miR-221 displayed a positive correlation with ferritin. Both neutrophil count and ferritin are crucial markers of the immune response. Pathway analysis revealed that miR-146a, miR-155, and miR-221 were significantly associated with both innate and adaptive immune responses (*Gaytan-Pacheco et al., 2022*).

The significance of miRNAs in shaping immune responses after COVID-19 vaccination, particularly with inactivated vaccines, remains poorly understood. In a study of cancer patients who received the mRNA COVID-19 booster vaccine, the levels of miR-7-5p, miR-15b-5p, miR-24-3p, and miR-223-3p in plasma extracellular vesicles increased significantly 6 months post-vaccination. These miRNAs were also positively correlated with anti-spike antibody levels, highlighting their role in regulating antiviral responses and cytokine production (*Almeida et al., 2024*). In pregnant women received three doses of mRNA COVID-19 vaccine, seven plasma miRNAs including miR-1972, miR-191-5p, miR-423-5p; miR-16-5p, miR-486-5p, miR-21-5p, and miR-451a exhibited different expression compared to unvaccinated pregnant women (*Lin et al., 2022*). It has been postulated that miRNAs may play a crucial role in myocarditis, arising from the heightened immune response elicited by mRNA COVID-19 vaccines (*AbdelMassih et al., 2022*). miR-92a-2-5p expression from circulating EVs before vaccination was negatively associated with adverse reactions after mRNA vaccine for COVID-19, while miR-148a expression was correlated with antibody production (*Miyashita et al., 2022*). In this study, blood samples were obtained from volunteers before and after a homologous booster (third) dose of the inactivated vaccine CoronaVac. Differential expression of miRNAs in peripheral blood mononuclear cells (PBMCs) was examined *via* microarray and qRT-PCR.

## MATERIALS AND METHODS

### Human study subjects

There were six and 31 volunteers in the discovery and validation cohorts, respectively. This prospective observational study was conducted during the government-launched COVID-19 vaccination campaign in 2021. Thirty-seven health care workers from Children's Hospital of Zhejiang University School of Medicine were enrolled into the study. Each of the participant had received two doses of CoronaVac with an interval of 28 days between February 23, 2021 and May 28, 2021. Subjects who had previously tested positive for SARS-CoV-2 were excluded from the study. The study protocol has received the ethics committee approval from the Children's Hospital of Zhejiang University School of Medicine (EC/IRB approval number: 2021029). Our study was in compliance with the recommendations detailed in the Declaration of Helsinki for biomedical research. Enrolled subjects provided written informed consent and received a third homogeneous dose of

CoronaVac at least 6 months after the second dose between November 16, 2021 and January 18, 2022. Peripheral blood samples were obtained from the subjects before and 1–2 weeks after the booster vaccination.

## Isolation of PBMCs

Peripheral blood samples (15–20 ml) were transferred into heparin tubes and processed within 4 h of collection. The blood was mixed thoroughly with an equal volume of phosphate-buffered saline (PBS). The diluted samples were then carefully placed on top of Lymphocyte Separation Medium (TBD, Tianjin, China) and centrifuged at 500 $g$ for 25 min at 20 °C to form the PBMC layer. PBMCs were aspirated from the centrifugation tubes using sterile glass pipettes, diluted with 4× volume of PBS, and harvested by centrifugation at 250 $g$ for 10 min. Cell pellets were resuspended with fetal bovine serum (FBS; Biological Industries, Kibbutz Beit-Haemek, Israel) supplemented with 10% DMSO (Sigma-Aldrich, Burlington, MA, USA). Then, cells ($1.5–3.0 \times 10^7$) were frozen in a stepwise manner by holding for 20 min at 4 °C, 1 h at −20 °C and overnight at −80 °C before moving into liquid nitrogen until usage.

## Microarray analysis

PBMCs samples from the discovery cohort (six each before and after vaccination) were thawed on ice, followed by centrifugation at 250 $g$ for 5 min. Then, total RNA was isolated from PBMC pellets with TRIzol (Thermo Fisher, Watham, MA, USA) and purified using the RNeasy Mini Kit (QIAGEN, Dusseldorf, Germany). The resulting RNA was assayed for quality and quantity by examining absorbance at 260 and 280 nm. miRNA labeling and hybridization were performed using a miRNA Complete Labeling and Hyb Kit (Agilent, Santa Clara, CA, USA). In brief, 500 ng of total RNA was labeled with cyanine 3 and hybridized to the SurePrint human G3 miRNA Microarray chip. After washing and drying, microarray images were captured using an Agilent Microarray Scanner. Array images were analyzed using Agilent Feature Extraction Software (v11.0.1.1). Agilent GeneSpring GX v12.1 software was employed for quantile normalization and data processing. To identify differentially expressed miRNA candidates, we defined two types of $p$-values: nominal $p$-value and adjusted $p$-value (false discovery rate approach). An adjusted $p$-value of <0.1, along with a fold-change of >1.5, was classified as differentially expressed. A heatmap analysis was conducted on the significantly upregulated and downregulated miRNAs, utilizing the SRplot web server (https://www.bioinformatics.com.cn/srplot).

## Real-time quantitative reverse transcriptase polymerase chain reaction (qRT-PCR)

Frozen PBMCs samples from the validation cohort (31 each before and after vaccination) were thawed on ice, followed by centrifugation at 250 $g$ for 5 min. Then, total RNA was isolated from PBMC pellets with TRIzol (Thermo Fisher, Watham, MA, USA), treated with DNase I, and purified using the RNeasy Mini Kit (QIAGEN, Dusseldorf, Germany). The quality and quantity of total RNA was assayed by examining absorbance at 260 and 280 nm using Nanodrop 2000 spectrophotometer (Thermo Fisher, Waltham, MA, USA).

The miRNA was reversely transcribed into cDNA using Mix-X miRNA First-Strand Synthesis kit (Takara Bio, Kusatsu, Japan) from 1 μg total RNA and 0.5 μl 100 nM primer using a T100 Thermo Cycler (Bio-Rad, Hercules, CA, USA) under condition of 1 h at 37 °C and 5 min at 85 °C. Samples were assayed in duplicate with both no-template controls and no-reverse transcriptase controls. Quantitative real-time PCR step was carried out with TB Green Premix Ex Taq II (Tli RNase H Plus) kit (Takara Bio, Shiga, Japan). The reaction was conducted under cycling condition of 10 s at 95 °C, and 40 cycles of 5 s at 95 °C and 20 s at 60 °C using LightCycler 480 Instrument II (Roche, Basel, Switzerland). To minimize inter-sample variability, all samples were processed simultaneously by a single individual, ensuring a consistent procedure and timing at each step to achieve uniform RNA quality and integrity. In addition, all samples for the same miRNA were analyzed in the same PCR run to ensure consistent amplification efficiency. Levels of all miRNAs were analyzed using LightCycler 480 software and normalized with U6 snRNA provided in the Mix-X miRNA First-Strand Synthesis kit. Relative miRNA expression were calculated using the ΔΔCt method (*Davoodian et al., 2014*). Primers for miR-25-5p, miR-299-5p, miR-129-5p, miR-206, miR-34c-3p, miR-494-3p were acquired from Genecopoeia (Rockville, MD, USA). The sequence for human snRNA U6 was 5′ TCGTGAAGCGTTCCATATTTTTAA3′ (Takara Bio, Kusatsu, Japan).

## Measurement of receptor binding domain (RBD)-specific IgG in plasma

ELISA plates (42592; Corning, Corning, NY, USA) were coated overnight at 4 °C with 1 μg/ml of RBD protein (Z03483; Genscript, Piscataway, NJ, USA) in 0.05 M carbonate-bicarbonate buffer, pH 9.4 and incubated with blocking buffer (CNB0011; Thermo Fisher, Waltham, MA, USA) for 1 h. After discarding the solution and rinsing with washing buffer (CNB0011; Thermo Fisher, Waltham, MA, USA) 3 times, diluted plasma was added to each well. After 3 times washing, a rabbit anti-human IgG-HRP antibody (ab6759; Abcam, Cambridge, UK) was added to the plate for 1 h. Plates were subsequently developed with 3,3′,5,5′-tetramethylbenzidine (TMB) substrate for 25 min. Then, hydrochloric acid (2 M) was added to stop the reaction. Optical density (OD) at 450 nm was measured *via* a microplate reader (SpectraMAX 190; Molecular Device, San Jose, CA, USA).

## Statistical analysis

Statistical analysis was performed using GraphPad Prism 8.0.2 software. Mann–Whitney test was used to compare miRNA expression before and after booster dose. Data were presented as median and interquartile range (IQR). Correlations were examined by the Spearman's correlation coefficient. The *p*-value $< 0.05$ was regarded as statistically significant.

## Bioinformatic analysis

Computationally predicted gene targets were acquired using miRDB and TargetScan databases. TargetScan combines thermodynamic modeling of miRNA-mRNA interactions with comparative sequence analysis, facilitating the accurate prediction of conserved

**Table 1  Demographic and characteristics of vaccinees.**

|  | Validation cohort | Discovery cohort | p-value |
|---|---|---|---|
| Vaccinees (n) | 31 | 6 | |
| Age, years (IQR) | 31 (28, 34) | 31 (30.5, 43.5) | >0.05 |
| Male/female (n) | 10/21 | 2/4 | >0.05 |
| Duration of post-vaccination, day (IQR) | 12 (11, 13) | 12 (10.5, 13) | >0.05 |

miRNA targets across diverse species (*Wang, 2008*). In contrast, miRDB focuses on mature miRNAs-the key players in miRNA functionality, offering both target prediction and functional annotation (*Wang, 2008*). The integration of miRDB and TargetScan allows the enhanced performance of target analysis and reduces the likelihood of identifying false target genes (*Oliveira et al., 2017*). To explore the signal pathways regulated by these target genes, KEGG pathway enrichment analysis (https://www.kegg.jp) was performed using KOBAS version 3.0 software. Pathways with nominal $p$-value < 0.01 and adjusted $p$-value < 0.05 were deemed as statistically significant (Fisher's exact test). Experimentally validated miRNA-gene interactions were retrieved from DIANA-TarBase v.8.

## RESULTS

### Study subject characteristics

The characteristics of volunteers for the present study are displayed in Table 1. The discovery cohort enrolled six healthcare workers. Concurrently, 31 healthcare workers were enrolled into the validation cohort. The interval between the second and third homogeneous doses of CoronaVac was at least 6 months. The booster dose had only mild local and systemic adverse events.

### Levels of miR-25-5p, miR-34c-3p, and miR-206 are altered after the booster dose

In the discovery cohort, a miRNA microarray of PBMCs obtained before and 1–2 weeks after the booster dose identified a sum of 1,791 miRNAs. The nominal and normalized array data was deposited in the Gene Expression Omnibus repository, with accession number GSE249050. Thirty of the uncovered miRNAs were upregulated by more than 1.5-fold (nominal $p$-value < 0.05), whereas 37 miRNAs had greater than 1.5 fold downregulation (Fig. 1A). Figure 1B showed the differential expression of miRNAs as a Volcano plot. After adjustment of the $p$-values, 26 miRNAs were classified as significantly upregulated (adjusted $p$-value < 0.1), while 36 miRNAs remained significantly downregulated (File S1).

We choose to focus on six miRNAs including miR-25-5p (nominal $p$-value = 0.029, adjusted $p$-value = 0.068), miR-34c-3p (nominal $p$-value = 0.031, adjusted $p$-value = 0.051), miR-129-5p (nominal $p$-value = 0.046, adjusted $p$-value = 0.049), miR-494-3p (nominal $p$-value = 0.021, adjusted $p$-value = 0.071), miR-299-5p (nominal $p$-value= 0.038, adjusted $p$-value= 0.048), and miR-206 (nominal $p$-value = 0.031, adjusted $p$-value = 0.048). The six

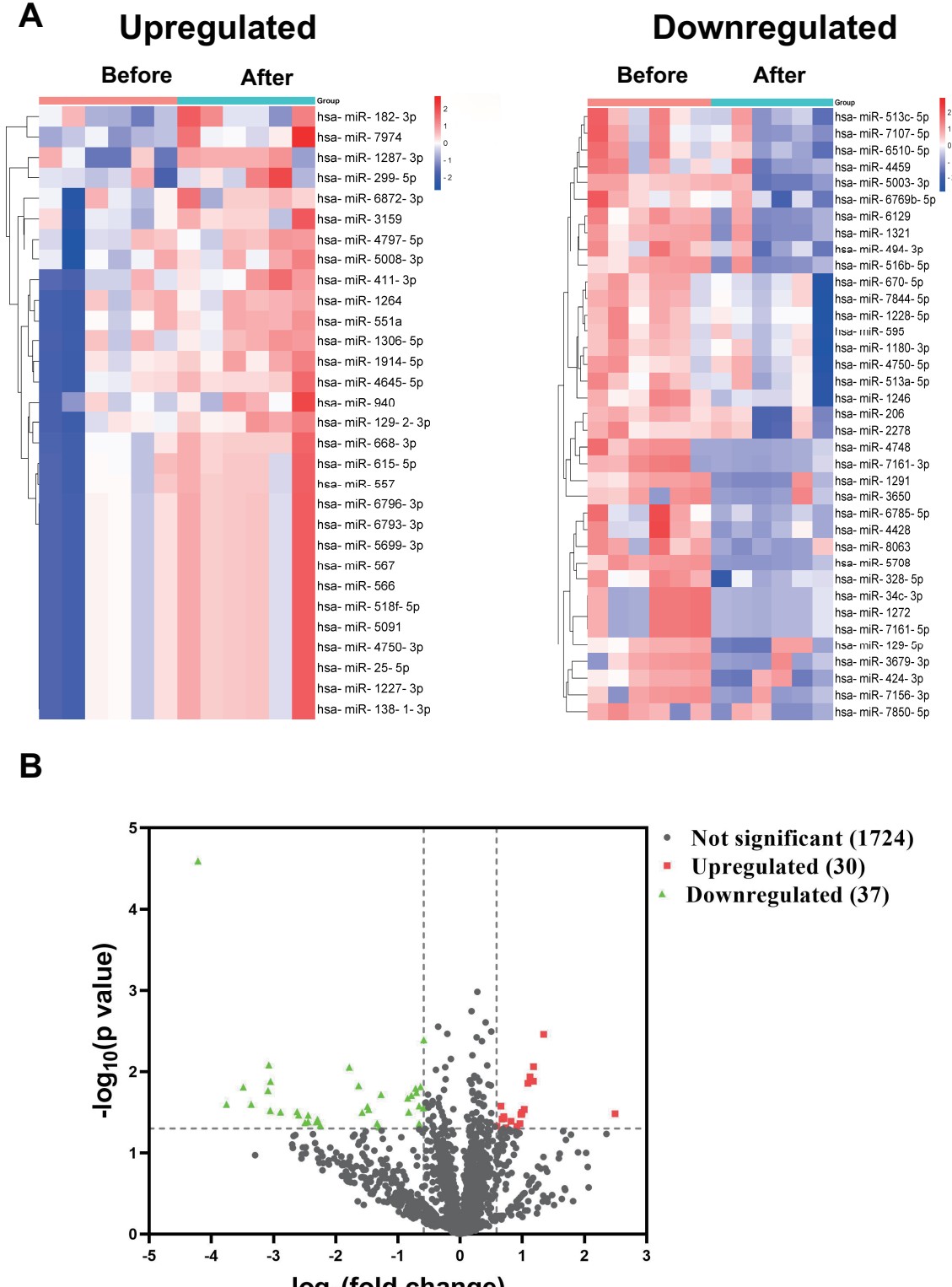

**Figure 1 miRNA profile of PBMCs in the discovery cohort before and after the booster dose of CoronaVac.** (A) A miRNA microarray was performed on PBMCs from six volunteers before and 1–2 weeks after the booster dose. There were 30 upregulated miRNAs and 37 downregulated miRNAs (greater than 1.5 fold, nominal $p < 0.05$). (B) The volcano plot illustrates the identification of differentially expressed miRNAs. Red dots represent miRNAs with a nominal $p$-value $< 0.05$ and a fold change $> 1.5$, while green dots indicate miRNAs with a nominal $p$-value $< 0.05$ and a fold change $< -1.5$. Black dots show 1,724 unaltered miRNAs.                     

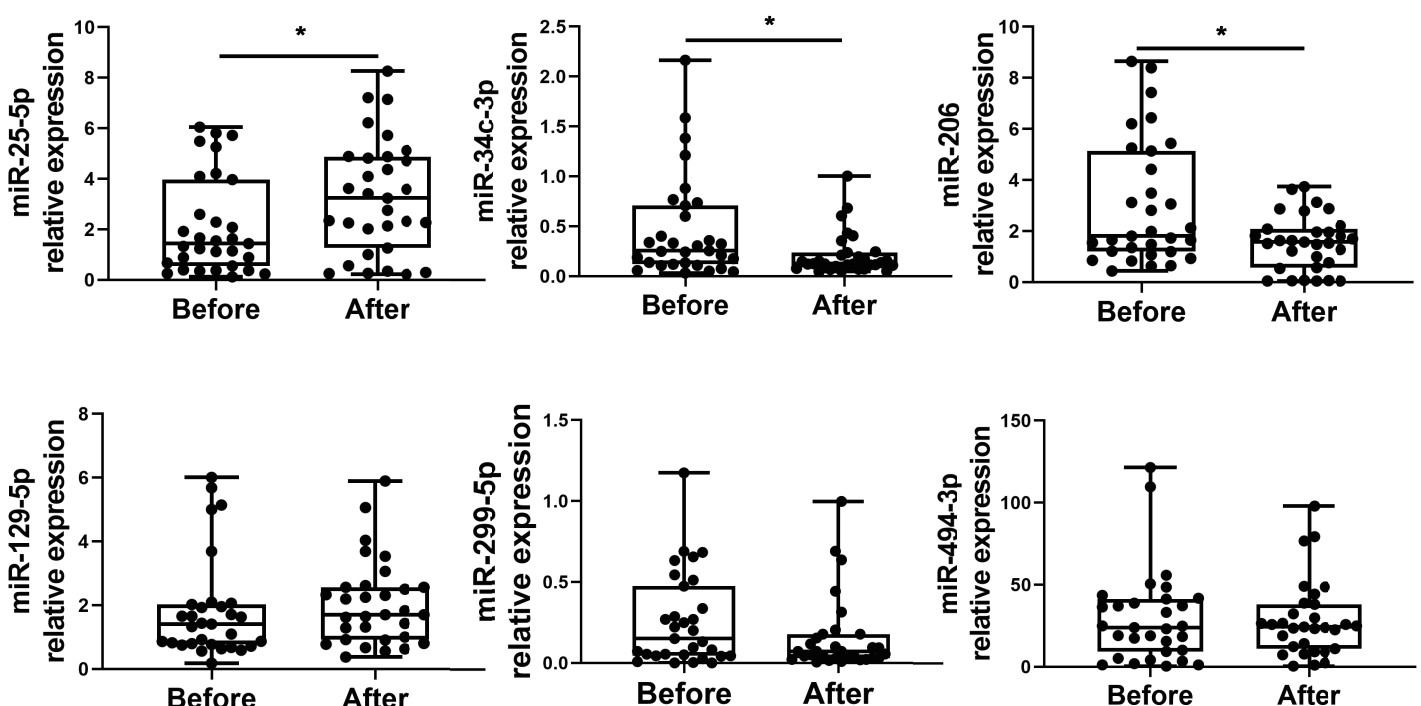

**Figure 2 Altered expression of miR-25-5p, miR-34c-3p, and miR-206 after the booster dose of CoronaVac in the validation cohort.** qRT-PCR was performed on PBMCs from 31 volunteers for the expression of six immune response-related miRNAs with differential expression. *n* = 31. The plot shows a median (lines within boxes), interquartile range (bounds of boxes), and error bars (upper and lower ranges). *$p < 0.05$.

selected miRNAs are known to play significant roles in immune responses (*Wu et al., 2021c*; *Tambyah et al., 2013*; *Song et al., 2015*; *Biswas et al., 2019*; *Xu et al., 2017*; *Akula, Bolin & Cook, 2022*). In contrast, the roles of the other miRNAs in regulating immune responses remain uncertain, as they have not been as extensively studied or significantly associated with immune mechanisms in the current literature. Subsequently, PBMCs from the validation cohort of 31 healthcare workers were examined to determine the levels of the six immune response-related miRNAs. Levels of miR-25-5p (adjusted *p*-value = 0.032) were significantly elevated after the booster dose as determined by qRT-PCR, while miR-34c-3p (adjusted *p*-value = 0.016) and miR-206 (adjusted *p*-value = 0.003) were significantly downregulated, with age and sex accounted for as covariates. However, there was no significant change in the expression of miR-129-5p (adjusted *p*-value = 0.520), miR-494-3p (adjusted *p*-value = 0.826), and miR-299-5p (adjusted *p*-value = 0.098) (Fig. 2) (File S1).

## miR-25-5p, miR-34c-3p, and miR-206 expression after booster dose is correlated with the production of RBD-specific IgG

Levels of RBD-specific IgG in plasma as reflected by OD450 values were significantly increased at 1–2 weeks after the booster dose (Fig. 3A). Spearman's correlation analysis was performed between the levels of miR-25-5p, miR-34c-3p, and miR-206 and the OD450 values of RBD-specific IgG. The analysis showed that expression of miR-25-5p

**A**

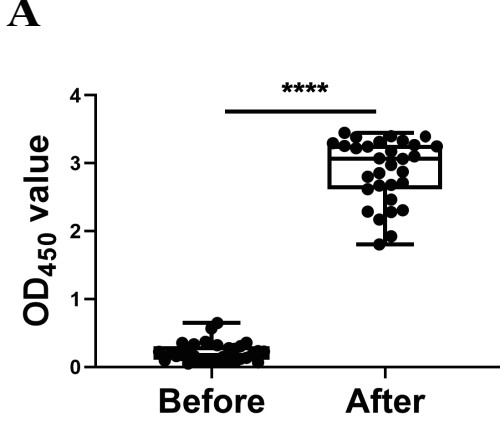

**B**

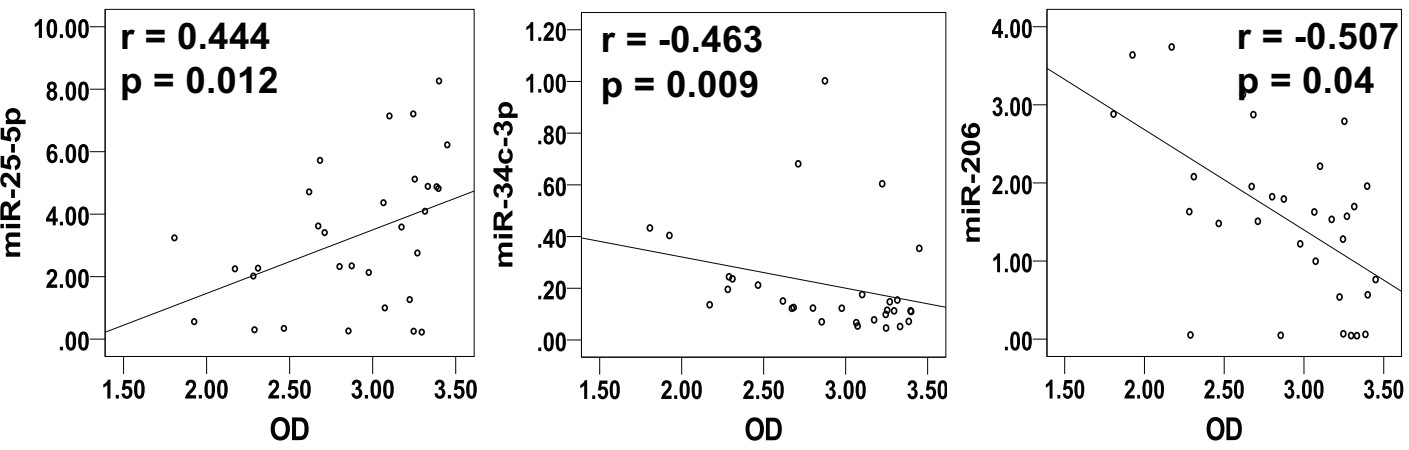

**Figure 3 Correlation between expression of miRNAs and levels of anti-RBD IgG.** (A) Levels of anti-RBD IgG were determined *via* ELISA and represented by $OD_{450}$ value. Each dot represents a subject. The plot shows a median (lines within boxes), interquartile range (bounds of boxes), and error bars (upper and lower ranges). ****$p < 0.001$. (B) The curve was plotted by the relative expression of miR-25-5p, miR-34c-3p, and miR-206 after booster dose to their respective $OD_{450}$ value of anti-RBD IgG. OD, Optical density.

($r = 0.444$, $p = 0.012$) after booster dose was positively correlated with the production of anti-RBD IgG, while miR-34c-3p ($r = -0.463$, $p = 0.009$) and miR-206 ($r = -0.507$, $p = 0.04$) were negatively associated with anti-RBD IgG (Fig. 3B). Conversely, miR-129-5p, miR-299-5p, and miR-494-3p were not correlated with the levels of anti-RBD IgG (data not shown). Collectively, these results suggest that miR-25-5p, miR-34c-3p, and miR-206 are associated with CoronaVac-induced immune responses.

### miR-25-5p, miR-34c-3p, and miR-206 regulate immune responses *via* multiple pathways

Forty-five overlapping target genes of upregulated miR-25-5p were identified using the online tools miRDB and TargetScan. In the meantime, 334 genes were recognized as target genes for downregulated miR-34c-3p, while 127 genes were perceived as target genes for

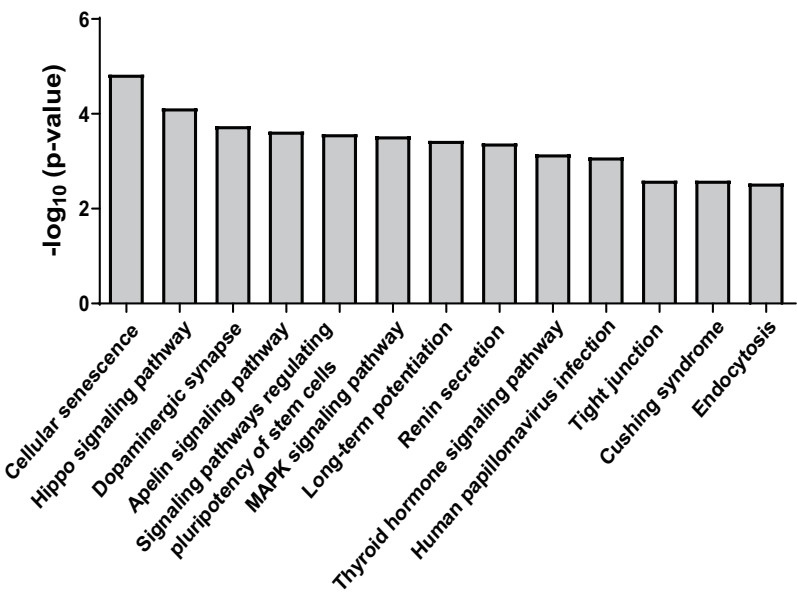

**Figure 4 KEGG pathways significantly enriched in target genes of miR-25-5p, miR-34c-3p, and miR-206.** KEGG pathway analysis was conducted for miR-25-5p, miR-34c-3p, and miR-206. There were 13 over-represented biological pathways for mRNA targets of the three miRNAs. Data shown are the negative log *p*-value (nominal) of the Fisher's exact test.

down-regulated miR-206. To explore the potential consequences of the interactions between the three miRNAs and their predicted target genes, KEGG pathway enrichment analysis was performed by importing the predicted target genes. Overall, 13 non-cancer pathways were predicted to be significantly modulated by miR-25-5p, miR-34c-3p, and miR-206 (nominal *p*-value < 0.01, adjusted *p*-value < 0.05) (Fig. 4) (Table 2). These pathways include Endocytosis (viral entry and innate immune responses to viral infection), Hippo signaling pathway (innate and adaptive immunity), MAPK signaling pathway (inflammation, immune cell differentiation, and immune cell proliferation), human papillomavirus infection (innate, humoral, and cellular immunity), and Tight junction (effector and target of immune regulation).

## miR-25-5p, miR-34c-3p, and miR-206 are associated with target genes for immune response

To further ascertain the relevance of miR-25-5p, miR-34c-3p, and miR-206 with immune response, we reviewed the overlapping target genes retrieved from TargetScan and miRDB. miR-25-5p, miR-34c-3p, and miR-206 were linked to 28 mRNAs in pathways highly relevant to immune functions. The majority of the interactions between miRNA and mRNAs was predicted, with two experimentally validated (Fig. 5). Among the 28 target mRNAs, six genes are linked to human papillomavirus infection, while three genes play a role in endocytosis. Ten genes belong to the MAPK pathway, while seven other genes are associated with the Hippo signaling pathway. In addition, two target genes (AMOT and PRKCE) modulate the function of tight junction, while two other genes (CDC42 and PRKACB) participate in both human papillomavirus infection and tight junction (File S1).

**Table 2  KEGG pathways for target genes of miR-25-5p, miR-34c-3p, and miR-206.**

| miRNA | Pathway | $p$-value | Adjusted $p$-value |
|---|---|---|---|
| miR-34c-3p | Cellular senescence | 1.46E−05 | 3.26E−03 |
| miR-34c-3p | Hippo signaling pathway | 7.42E−05 | 8.31E−03 |
| miR-34c-3p | Dopaminergic synapse | 1.77E−04 | 1.08E−02 |
| miR-34c-3p | Apelin signaling pathway | 2.30E−04 | 1.08E−02 |
| miR-34c-3p | Signaling pathways regulating pluripotency of stem cells | 2.61E−04 | 1.08E−02 |
| miR-34c-3p | MAPK signaling pathway | 2.90E−04 | 1.08E−02 |
| miR-34c-3p | Long-term potentiation | 3.61E−04 | 1.15E−02 |
| miR-34c-3p | Renin secretion | 4.10E−04 | 1.15E−02 |
| miR-34c-3p | Cushing syndrome | 2.51E−03 | 2.81E−02 |
| miR-206 | Thyroid hormone signaling pathway | 6.97E−04 | 2.84E−02 |
| miR-206 | Human papillomavirus infection | 8.04E−04 | 2.84E−02 |
| miR-206 | Tight junction | 2.50E−03 | 4.49E−02 |
| miR-25-5p | Endocytosis | 2.87E−03 | 1.55E−02 |

# DISCUSSION

miRNAs have been recognized as important biomarkers of immune response to vaccination (*Visacri et al., 2021*). However, the effects of COVID-19 vaccination on mRNA profiles in immune cells are poorly documented. The present study examined miRNA profile in PBMCs after a homologous booster dose with inactivated SARS-CoV-2 vaccine. Our results showed that there was differential expression of miR-25-5p, miR-34c-3p, and miR-206 after the booster dose. miR-25-5p, miR-34c-3p, and miR-206 were correlated with the production of anti-RBD IgG. These miRNAs may target multiple pathways involving the immune response.

miRNAs control gene expression at the post-transcriptional level and are key regulators in the innate and adaptive immune response. miRNAs have been demonstrated as biomarkers for lymphocyte activation and infectious diseases (*Correia et al., 2017*; *de Candia et al., 2014*). Circulating miRNAs have been widely explored as biomarkers for infectious diseases and immune responses. However, they are not generated directly from cytokine-producing immune cells. Many cell types including neutrophils, PBMCs, platelets, and endothelial cells are able to generate circulating miRNAs (*Pan et al., 2014*; *Mittelbrunn et al., 2011*). PBMC miRNA expression profile after the booster dose of CoronaVac was examined in the present study. Several previous studies have documented the differential expression of vaccine-induced miRNAs in PBMCs. *Sailo et al. (2019)* reported that miR-22-5p and miR-27b-5p were differentially expressed in PBMCs after vaccination for classical swine fever. miR-146a, miR-326, and miR-155 were downregulated in PBMCs of BCG-vaccinated subjects (*Corral-Fernandez et al., 2016*). In sheep inoculated with vaccine for Peste Des Petits Ruminants Virus, miR-150, miR-370-3p, and miR-411b-3p were differentially expressed in PBMCs (*Yang et al., 2019*). In lambs that received vaccine adjuvant of aluminum hydroxide, six miRNAs were

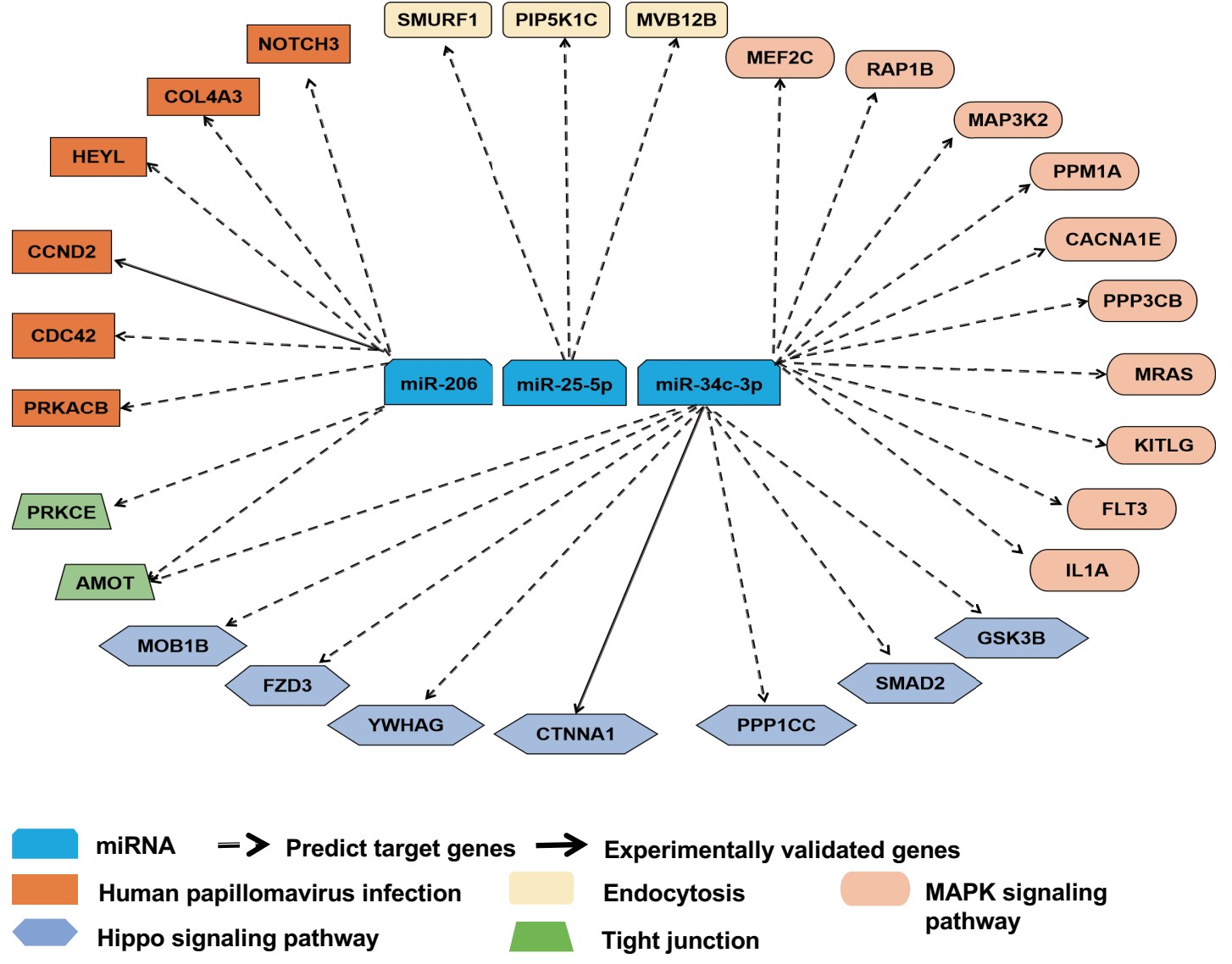

**Figure 5 Target gene network for miR-25-5p, miR-34c-3p, and miR-206 with relevance to immune response.** Solid and dashed arrows show experimentally validated and predicted targets, respectively.

differentially expressed in PBMCs (*Varela-Martinez et al., 2018*). The present study may provide new insights into the mechanisms of inactivated SARS-CoV-2 vaccine.

miR-25-5p, miR-34c-3p, and miR-206 have been reported as potential markers for COVID-19 and regulators of the immune response. *Arisan et al. (2022)* discovered that miR-34c-3p expression was downregulated in Vero cells infected with SARS-CoV-2. Following treatment of anti-miR-1307, a critical miRNA in SARS-CoV-2 infection, the expression of miR-34c-3p was significantly downregulated (*Arisan et al., 2022*). miR-34c-3p was among the most significant differentially expressed circulating miRNAs in COVID-19 patients requiring mechanical ventilation compared with patients without mechanical ventilation requirement (*Garcia-Giralt et al., 2022*). In circulating EVs, miR-206 was

upregulated in COVID-19 patients with acute respiratory distress syndromes *vs* with pneumonia (*Meidert et al., 2021*). In the early stage of SARS-CoV-2 infection of Vero cells, miR-206 was significantly downregulated (*Liu et al., 2022a*). Upregulation of miR-206 and miR-205-5p upon hospital admission was shown in critical COVID-19 compared with non-critical survivors, suggesting an association between miR-206 and disease severity (*Vaz et al., 2023*). miR-25-5p was identified as one of the miRNAs that could potentially interact with the genome of SARS-CoV-2 (*Milenkovic et al., 2021*). It has been reported that natural infection of COVID-19 and vaccination both produce a similar immune response (*Samanovic et al., 2022*). Therefore, miR-25-5p, miR-34c-3p, and miR-206 may have the potential to be used as biomarkers for both COVID-19 infection and vaccination.

In the present study, miR-25-5p, miR-34c-3p, and miR-206 were associated with 28 target genes involved in immune response. One of the predicted target genes for miR-25-5p is PIP5K1C (Fig. 5) which plays a role in the regulation of endocytosis pathway (*Roy et al., 2023*). Inhibition of PIP5K1C blocked ACE2-mediated endocytosis of SARS-CoV-2 virus into host cells (*Seo et al., 2024*). In addition, integrin induced PIP5K1C polarization and subsequent neutrophil infiltration (*Xu et al., 2010*). One of our predicted targets of miR-34c-3p is MAP3k2 in the MAPK pathway, which was involved in T-cell receptor signaling to activate JNK and elevate IL-2 expression (*Su et al., 2001*). The second potential target of miR-34c-3p is PPP3CB (calcineurin A beta), a player in MAPK pathway. In mice with PPP3CB knockout, there was a defect in T cell development and function (*Bueno et al., 2002*). In addition, PPM1A, also part of the MAPK pathway, is a prospective target of miR-34c-3p. PPM1A balanced antiviral signal transduction *via* dephosphorylating STING and TBK1, two regulators for type I interferon production (*Li et al., 2015*). Furthermore, SMAD2 is a potential target of miR-34c-3p and a mediator of TGF-β signaling. SMAD2 was shown to crosstalk with Hippo signaling pathway (*Ghomlaghi et al., 2024*). *Garcia et al. (2022)* reported that Hippo pathway activation had antiviral function in SARS-CoV-2 infection. NOTCH3, a component in the human papillomavirus pathway, is our predicted target gene for miR-206. SARS-CoV-2 infection was revealed to activate a variant of NOTCH3, causing the development of cerebral autosomal dominant arteriopathy with subcortical infarcts and leukoencephalopathy (CADASIL) (*Krol et al., 2023*). CDC42, also a player in human papillomavirus pathway, is a potential target of miR-206. Equine herpesvirus type 1 activated CDC42 which stabilized tubulin and enhanced the intracellular transport of virus and spread to adjacent cells (*Kolyvushko et al., 2020*). In a study of a group of healthcare workers who received two doses of CoronaVac, *Chen et al. (2023)* found that MAPK1, CDC42, PPP2AC, EP300, YWHAZ, and NRAS were correlated with the vaccine response. Among them, CDC42 is identified as a potential target gene of miR-206 in our prediction, while PPP2CA and YWHAZ closely resemble our predicted miR-34c-3p targets, PPP3CB and YWHAG, respectively.

The present study suggests that miR-25-5p, miR-34c-3p, and miR-206 in PBMCs are potential biomarker candidates for humoral response after the third dose of CoronaVac. Expression of miR-25-5p, miR-34c-3p, and miR-206 has also been associated with immune response in non-COVID-19 studies. *Wu et al. (2021c)* reported that Peg-IFN-α

treatment enhanced miR-25-5p levels in circulating EVs of patients with hepatitis B virus, resulting in inhibition of virus replication and transcription. *Wang et al. (2022)* found that miR-25-5p alleviated lipopolysaccharide (LPS)-elicited inflammatory response, production of reactive oxygen species, and brain damage *via* targeting TXNIP. *Lu et al. (2016)* revealed that miR-25-5p elevated M2 macrophages *via* inhibiting multiple genes in NF-κB and MAPK signaling pathways. *Haidar et al. (2023)* showed that there was differential expression of miR-34c-3p in *T. annulata*-infected leukocytes. *Peng et al. (2016)* revealed that miR-34c-3p from throat swab had the highest diagnostic value for influenza infection. *Wright et al. (2021)* found that mycobacterial infection enhanced the expression of miR-206, which inhibited neutrophil recruitment to the infected site. Lastly, *Liu et al. (2022b)* reported that miR-206 induced M1 polarization of Huffer cells and hepatic recruitment of $CD8^+$ T cells.

Both miRNAs and RBD-specific antibody may serve as valuable biomarkers for assessing the immune response following COVID-19 vaccination. One advantage of miRNA biomarkers is their potential for early detection, making them useful as initial indicators of immune response. However, a significant drawback is their lack of specificity in responding to various stimuli such as infections and vaccinations. For instance, miR-155 has been shown to regulate immune responses in both sepsis and COVID-19 (*Papadopoulos, Papadopoulou & Aw, 2023*), while changes in miR-146 may reflect a more generalized immune activation through the NF-κB pathway (*Testa et al., 2017*). In contrast, RBD-specific antibody targets the receptor-binding domain of the virus, which mediates viral entry into host cells (*Miyashita et al., 2022*). This specificity allows RBD-specific antibody to provide a more accurate assessment of the body's immune response to the virus. Therefore, RBD-specific antibody may represent a stronger and more clinically relevant marker than altered miRNA profile alone. Nevertheless, the combination of both miRNAs and RBD-specific antibody provides a comprehensive perspective on vaccine efficacy and overall immunity.

The present study demonstrated that expression of miR-25-5p, miR-34c-3p, and miR-206 in PBMCs is correlated with the production of RBD-specific IgG, which has been associated with neutralizing activity and virus control (*Wu et al., 2021a*). Additionally, during the acute phase of COVID-19 infection, levels of plasma miR-497-5p also exhibited a correlation with the RBD-specific IgG (*Wu et al., 2021b*). However, the correlation does not imply causation, nor does it confirm that changes in the miRNAs levels directly affect antibody production. Firstly, multiple miRNAs can regulate overlapping immune pathways, making it hard to pinpoint the specific roles of each miRNA in the immune response. This overlap may lead to spurious correlations, potentially obscuring genuine relationships and the roles of miRNAs in regulating the humoral response. Secondly, validation experiments are essential to identify the target genes and downstream pathways that mediate the observed alterations in RBD-specific antibody.

The present study has several limitations. First, while we observed the potential of miR-25-5p, miR-34c-3p, and miR-206 as biomarkers of CoronaVac-induced immunity, we may not assert that these findings are universally applicable due to the low statistical power

stemming from our small sample size. This constraint may have also impeded our ability to detect genuine alterations in the miRNA profile. Consequently, the miRNA biomarker data related to the CoronaVac-induced immune response should be regarded as preliminary and warrants further validation with larger cohorts. Second, the time point for examining miRNA expression was performed at 1–2 weeks after booster dose. The durability of altered miRNA expression profile warrants further investigation. Third, this study is limited to *in silico* prediction of target genes and future experiments are needed to validate these target genes.

## CONCLUSIONS

The present study suggests the potential role of miR-25-5p, miR-34c-3p, and miR-206 as biomarkers for CoronaVac-induced immune response. Future research should focus on larger-scale studies with longer-term follow-up to validate the preliminary findings and identify pathways that mediate the immune response.

### Funding

This work was supported by the Natural Science Foundation of Zhejiang Province (Grant No. LGF22H150010), the National Natural Science Foundation of China (Grant No. 82070074, 82272191, and 82370080), the Fundamental Research Funds for the Central Universities (Grant No. 226202200060), and the Key R&D Program of Zhejiang (Grant No. 2022C03163), the Health Commission of Zhejiang Province (Grant No. 2024KY491), and the Health Science Commission of Shaoxing (Grant No. 2023SKY 100 and 2023SKY 103). The funders had no role in study design, data collection and analysis, decision to publish, or preparation of the manuscript.

### Grant Disclosures

The following grant information was disclosed by the authors:
Natural Science Foundation of Zhejiang Province: LGF22H150010.
National Natural Science Foundation of China: 82070074, 82272191, and 82370080.
Fundamental Research Funds for the Central Universities: 226202200060.
Key R&D Program of Zhejiang: 2022C03163.
Health Commission of Zhejiang Province: 2024KY491.
Health Science Commission of Shaoxing: 2023SKY 100 and 2023SKY 103.

### Competing Interests

The authors declare that they have no competing interests.

### Author Contributions

- Guanguan Qiu conceived and designed the experiments, performed the experiments, analyzed the data, prepared figures and/or tables, authored or reviewed drafts of the article, and approved the final draft.

- Ruoyang Zhang conceived and designed the experiments, performed the experiments, analyzed the data, prepared figures and/or tables, authored or reviewed drafts of the article, and approved the final draft.
- Huifeng Qian conceived and designed the experiments, performed the experiments, analyzed the data, prepared figures and/or tables, authored or reviewed drafts of the article, and approved the final draft.
- Ruoqiong Huang conceived and designed the experiments, performed the experiments, analyzed the data, prepared figures and/or tables, authored or reviewed drafts of the article, and approved the final draft.
- Jie Xia conceived and designed the experiments, performed the experiments, analyzed the data, prepared figures and/or tables, authored or reviewed drafts of the article, and approved the final draft.
- Ruoxi Zang conceived and designed the experiments, performed the experiments, analyzed the data, prepared figures and/or tables, authored or reviewed drafts of the article, and approved the final draft.
- Zhenkai Le conceived and designed the experiments, performed the experiments, analyzed the data, prepared figures and/or tables, authored or reviewed drafts of the article, and approved the final draft.
- Qiang Shu conceived and designed the experiments, analyzed the data, prepared figures and/or tables, authored or reviewed drafts of the article, and approved the final draft.
- Jianguo Xu conceived and designed the experiments, analyzed the data, prepared figures and/or tables, authored or reviewed drafts of the article, and approved the final draft.
- Guoping Zheng conceived and designed the experiments, analyzed the data, prepared figures and/or tables, authored or reviewed drafts of the article, and approved the final draft.
- Jiangmei Wang conceived and designed the experiments, performed the experiments, analyzed the data, prepared figures and/or tables, authored or reviewed drafts of the article, and approved the final draft.

## Human Ethics

The following information was supplied relating to ethical approvals (*i.e.*, approving body and any reference numbers):

The study protocol has received the ethics committee approval from the Children's Hospital of Zhejiang University School of Medicine.

## Microarray Data Deposition

The following information was supplied regarding the deposition of microarray data:

The microarray data is available at GEO: GSE249050.

## Data Availability

The raw data is available in the Supplemental File.

## Supplemental Information

Supplemental information for this article can be found online at http://dx.doi.org/10.7717/peerj.18856#supplemental-information.

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
