# Peer review of "Altered expression of miRNA profile in peripheral blood mononuclear cells following the third dose of inactivated COVID-19 vaccine"

_PeerJ, doi:10.7717/peerj.18856_

## Round 0.1 · original submission · Major Revisions

Please address the concerns of all the reviewers and revise manuscript accordingly.

Reviewer 1 ·

Basic reporting

This manuscript delineates the expression of miRNA from homologous prime-boost of inactivated COVID-19 vaccine recipients on a pilot scale. The study focuses on the immune profiling of an inactivated COVID-19 that is lacking in existing literature, and this study may be one of the most interesting data in this scope. The findings and outcomes were sound and acceptable for publication. However, this manuscript is a pilot scale that has a limitation of a small sample size that may not reflect the large populations or other inactivated COVID-19 vaccines (non-CoronaVac).

Experimental design

'no comment'

Validity of the findings

The findings in this study are sound and could be replicable by the others.

Additional comments

Comments.
1. Abstract, lines 22-23: "have demonstrated that the third (booster) dose".
This statement may lead to confusion because it means a third dose (first booster vaccination) or a fifth dose (third booster vaccination). So, we suggest using "the third dose or booster vaccination" to clarify it.

2. Lines 40-44: "There are two main categories of vaccines for severe acute respiratory syndrome coronavirus 2 (SARS-CoV-2). The first category, which includes mRNA vaccines and adenovirus-vector vaccines, provides the nucleotide sequence for the synthesis of SARS-CoV-2 spike protein in the vaccinee. The second category, which consists of inactivated vaccines and subunit vaccines, provides protein antigen directly to the immune system [1]."
These statements seem unclear. Moreover, an inactivated vaccine platform is the first type of vaccine discovered by Edward Jenner. I think you should not classified as a "second" type.
Suggests rewriting it to make it clear, such as;
... The first category comprises antigens from viral particles or subunits, such as inactivated and protein subunit platforms. The second category uses genetic materials embedded in a specific nucleotide sequence to synthesise SARS-CoV2 antigens, such as mRNA and adenoviral vector platforms...

Anyways, it depends on your writing. My example statement is only a guide to make it more clear statements.

3. Table 1: This table should add more information on 6 volunteers who used a sample to perform miRNA (Figure 1) to make it more informative.
Because 6 volunteers may have a different central tendency (median with IQR) in age and duration of post-vaccination.

·

Basic reporting

Major Issues:
1. Clarity and Language: While the manuscript generally uses clear and professional language, there are a few areas where the wording could be improved for better clarity and flow. For example, the introduction could benefit from rephrasing to reduce redundancy and improve coherence. The authors sometimes repeat concepts (e.g., vaccination and immune response) without clearly linking them.
2. Introduction and Background: The introduction provides sufficient context about COVID-19 vaccines and the immune response, but it lacks depth in the explanation of the significance of miRNA in post-vaccine immune responses. This could be expanded to include more recent and relevant studies to strengthen the rationale for this study.
3. Lack of Code for Reproducibility: One significant issue is the absence of any code or detailed instructions for the bioinformatics analyses and figure generation. Since bioinformatics tools and scripts are critical for reproducing pathway analyses, differential expression analysis, and visualizations, the authors should provide the code or at least mention which specific software, versions, and parameters were used. This omission limits the reproducibility of the bioinformatics results and the figures, which is essential for transparency and validation by other researchers.

Minor Issues:
1. Terminology Consistency: Throughout the manuscript, there is some inconsistency in terminology. For instance, “CoronaVac” is sometimes referred to as the inactivated vaccine. It would be clearer to consistently refer to the vaccine as "CoronaVac" or at least define this early in the manuscript and stick to one term.
2. Literature Referencing: Although the manuscript references many relevant studies, some of the citations used are older. Incorporating more recent studies on COVID-19 and miRNA would strengthen the argument, especially those that focus on the specific role of miRNAs in COVID-19 immune responses.

Experimental design

Major Issues:
1. Sample Size: The study has a small sample size for both the discovery cohort (6 participants) and the validation cohort (31 participants). While the study does mention this as a limitation, this small sample size significantly affects the robustness and generalizability of the findings. Future studies should aim for a larger cohort size to confirm the miRNA biomarkers identified here.
2. Lack of Controls with No Vaccine: A significant concern in the experimental design is the absence of a control group that did not receive the vaccine. This makes it difficult to determine whether the changes in miRNA expression are specifically attributable to the vaccine or are due to other factors such as time-dependent changes in miRNA expression or environmental influences. Including a group that did not receive the vaccine, or alternatively, a control group receiving a placebo, would significantly strengthen the study’s conclusions about vaccine-induced changes in miRNA expression.
3. Time Points: The study examines miRNA expression at only one post-vaccination time point (1-2 weeks after the booster dose). This is a short period to capture potential long-term changes in miRNA expression that might be relevant to vaccine-induced immunity. More time points, including longer-term follow-ups (e.g., 6 months), would provide a more comprehensive picture of the changes in miRNA expression.
4. Details in Methodology: The methodology is generally described in sufficient detail, but some additional information is needed for reproducibility. For example, the section on qRT-PCR does not mention the controls used to ensure the accuracy of the measurements. Adding information on negative controls and how the authors mitigated inter-sample variability would improve confidence in the results.

Minor Issues:
1. Choice of miRNAs: The selection of 6 miRNAs for further validation is explained, but the rationale for focusing on these specific miRNAs should be more thoroughly justified in the text. While it is mentioned that these miRNAs are involved in immune responses, it would be useful to explain why they were chosen over other differentially expressed miRNAs.

Validity of the findings

Major Issues:
1. Statistical Analysis: The statistical analysis appears appropriate for the study, particularly the use of the Mann-Whitney test for comparing miRNA expression before and after vaccination. However, given the small sample size, the statistical power is likely limited. The manuscript should include a more detailed discussion of how this limitation affects the strength of the conclusions.
2. RBD-Specific Antibody Response as a Stronger Marker: The RBD-specific antibody response appears to be a more significant marker of immune protection compared to miRNA expression changes. The pronounced and statistically significant difference in RBD-specific antibody levels before and after the booster dose makes it a more direct and reliable measure of the vaccine-induced immune response. Since RBD antibodies directly target the virus’s receptor-binding domain, which is essential for viral entry into host cells, their levels provide a clearer indication of the body’s protective capacity. The manuscript should emphasize that RBD-specific antibody response may be a stronger and more clinically relevant marker than miRNA expression changes alone, although both markers together provide a comprehensive view of the immune response.
3. Correlation with Immune Response: The study correlates the expression of miR-25-5p, miR-34c-3p, and miR-206 with the receptor-binding domain (RBD)-specific antibody response. While this is a good finding, the authors should be cautious about claiming strong causal relationships based on correlation alone. The manuscript would benefit from a more nuanced discussion of the limitations of correlation-based findings and suggestions for future mechanistic studies to explore the functional roles of these miRNAs in vaccine responses.
4. Bioinformatics and Pathway Analysis: The bioinformatics analysis identifies pathways potentially regulated by the differentially expressed miRNAs. However, the interpretation of these findings is somewhat superficial. The discussion could benefit from a deeper exploration of the specific roles these pathways play in immune response and inflammation, especially in the context of vaccination.

Minor Issues:
1. Conclusion and Future Directions: The manuscript presents strong conclusions about the potential of miR-25-5p, miR-34c-3p, and miR-206 as biomarkers of vaccine-induced immunity. However, it would benefit from a more cautious tone given the small sample size and the limited time point investigated. Future directions should be more clearly outlined, especially regarding the need for larger studies and longer-term follow-up to validate these findings.
2. Data Presentation: In some cases, the data are presented without sufficient context. For example, Figure 3 shows correlations between miRNA expression and antibody levels, but these correlations are not well-explained in the text. More detailed explanations of the figures would improve reader comprehension.

Additional comments

The study provides investigation into the potential of miRNAs as biomarkers for immune responses following COVID-19 vaccination. However, the small sample size, limited time points, lack of reproducibility due to missing code, lack of in-depth mechanistic analysis and the lack of a control group significantly limit the strength of the conclusions. The manuscript would benefit from more detailed explanations in several areas, particularly in the methodology and the interpretation of results. Future studies should address these limitations to provide a more comprehensive understanding of miRNA responses to COVID-19 vaccination.

Reviewer 3 ·

Basic reporting

Request the authors to separately showcase the attributes of discovery (n=6) and validation set (n=31) individuals in Table 1 to rule out any skewness in selecting individuals for the study.

Experimental design

Request the authors to explain the rationale for choosing the two miRNA target databases.

Validity of the findings

It is critical to elaborate on the computational approach used to perform DE analysis of miRNA array data in the discovery cohort. The narrative seems to be loosely based on nominal significance of miRNAs discovered in 6 individuals.
In addition, only raw p-values are reported for the 6 miRNAs in the validation set (n=31). It is crucial for the authors to report adjusted p-values to make their claims robust.
Again figure 4 reports nominal p-values, request authors adapt their claims based on adjusted p-values.

Additional comments

Request the authors to provide accurate p-values and adjusted p-values rather than just stating the range.
Adding a volcano plot to figure 1 would permit the reader to gain more insights from the analysis.

---

## Round 0.2 · Major Revisions

Please address remaining concerns of the reviewers and amend manuscript accordingly.

Reviewer 1 ·

Basic reporting

Thank you for thoroughly addressing the concerns I raised in your previous submission.

Experimental design

'no comment'

Validity of the findings

The findings in this study are sound and could be replicable by the others.

·

Basic reporting

1) Clarity and Language Improvements: The introduction has been revised to eliminate redundancy and enhance clarity. Key phrases and terminology have been refined for consistency, notably ensuring "CoronaVac" is used systematically.
2) Background Expansion: Additional references (21-24) were added to elaborate on the role of miRNAs in immune responses, strengthening the study's rationale.
3) Terminology and Literature: Terminological inconsistencies were addressed, and newer studies were incorporated to solidify the scientific context.
4) Data Presentation: The inclusion of parameters for the 6 volunteer discovery cohort in Table 1, along with detailed figure descriptions, enhances readability and transparency.

Experimental design

1) Sample Size and Control Groups: While acknowledging the limitations of small sample size and the lack of an unvaccinated control group, the authors justified these constraints within the real-world healthcare context. These points were reflected in the revised discussion and conclusion.
2) Methodological Details: Bioinformatics tools, parameters, and additional procedural controls were explicitly outlined, enhancing reproducibility. For example, qRT-PCR controls were described in detail to mitigate inter-sample variability.
3) Target miRNA Rationale: The manuscript now includes justifications for selecting specific miRNAs based on their association with immune mechanisms, enhancing the scientific rationale.
4) Additional Enhancements: The authors addressed reviewer suggestions, such as providing a volcano plot in Figure 1 and reporting adjusted p-values alongside nominal values for robustness.

Validity of the findings

1) Statistical Analysis: Limitations due to small sample size were thoroughly discussed. Adjusted p-values for pathway analysis were added, with all significant findings remaining robust.
2) Correlation vs. Causation: The discussion now emphasizes the correlational nature of the findings and highlights the need for mechanistic studies to validate miRNA roles in immune responses.
3) RBD-Specific Antibody Response: The manuscript now provides a nuanced comparison of miRNA biomarkers and RBD-specific antibodies, underscoring their complementary roles in evaluating vaccine efficacy.
4) Pathway Analysis: The discussion of miRNA-regulated pathways in immune response was expanded with additional references, providing a deeper understanding of the findings.

Additional comments

The authors have made significant revisions that address many major and minor points raised earlier. These include improving clarity, enhancing methodological transparency, addressing limitations, and contextualizing findings within the broader scientific landscape. The manuscript has been substantially strengthened and is now ready for publication.

Reviewer 3 ·

Basic reporting

No comment

Experimental design

No comment

Validity of the findings

Major:
Appreciate the authors sharing the attributes of the discovery cohort in Table 1. It is not clear from the Microarray analysis section if the authors controlled for Age and Sex in the DE analysis. Can the authors confirm if the model used for DE analysis included Age and Sex as covariates? It is essential to control for any gender or age related effect given that there are 2 males and 4 females and two individuals (Number 1 and 2) are in a different age bracket.

Request the authors to adapt the outcome in the manuscript after re-analysis of discovery cohort including Age and Sex as covariates in the DE model if not already included.

Minor:
In the Microarray analysis section, authors have added the sentence, "A nominal p-value of < 0.05 or an adjust p-value of < 0.1, along with a fold-change of > 1.5, were classified as differentially expressed".

This sentence is confusing and urge the authors to report only FDR adjusted p-value since reporting DE miRNAs without adjustment is not meaningful. Given the low sample size, it is understandable that the FDR adjusted p-value cut-off is set to 0.1.
Appreciate the authors for having appropriately acknowledged the low statistical power in the study (Line 375-381).

Request the authors to report only adjusted p-values for DE analysis results addressing the discovery cohort. This includes changing the results section "Levels of miR-25-5p, miR-34c-3p, and miR-206 are altered after the booster dose" appropriately.

---

## Round 0.3 · accepted · Accept

All issues pointed by the reviewers were addressed, and revised manuscript is acceptable now.

Reviewer 3 ·

Basic reporting

No comment

Experimental design

No comment

Validity of the findings

No comment

Additional comments

The authors have duly caveated the low statistical power of the study and have taken some steps to account for nuisance variables given the constraints of low sample size.
Although this study does not significantly impact the current standard of clinical research, it has potential to serve as foundation for more structured and detailed research.